# Picornavirus May Be Linked to Parkinson’s Disease through Viral Antigen in Dopamine-Containing Neurons of Substantia Nigra

**DOI:** 10.3390/microorganisms10030599

**Published:** 2022-03-10

**Authors:** Bo Niklasson, Lars Lindquist, William Klitz, Sten Fredrikson, Roland Morgell, Reza Mohammadi, Yervand Karapetyan, Elisabet Englund

**Affiliations:** 1Jordbro Primary Health Care Center, 137 64 Stockholm, Sweden; roland.morgell@sll.se (R.M.); reza.mohammadi@sll.se (R.M.); 2Division of Infectious Disease, Department of Medicine Huddinge, Karolinska Institutet, 141 86 Stockholm, Sweden; lars.lindquist@ki.se; 3Department of Integrative Biology, University of California, Berkeley, CA 94720-3140, USA; klitzwilliam@gmail.com; 4Department of Neurology, Karolinska Institutet, Karolinska University Hospital, 141 86 Stockholm, Sweden; sten.fredrikson@ki.se; 5Netherlands Brain Bank, Netherlands Institute for Neuroscience, Meibergdreef 47, 1105 BA Amsterdam, The Netherlands; 6Integrated Biobank of Luxembourg 1, Rue Louis Rech, L-3555 Dudelange, Luxembourg; yervandkar@gmail.com; 7Division of Pathology, Department of Clinical Sciences, University of Lund, 221 85 Lund, Sweden; elisabet.englund@med.lu.se

**Keywords:** neurodegenerative disease, Parkinson’s disease, Ljungan virus, picornavirus, substantia nigra, immunohistochemistry

## Abstract

Parkinson’s disease (PD) is a neurodegenerative disease linked with the loss of dopaminergic neurons in the brain region called substantia nigra and caused by unknown pathogenic mechanisms. Two currently recognized prominent features of PD are an inflammatory response manifested by glial reaction and T-cell infiltration, as well as the presence of various toxic mediators derived from activated glial cells. PD or parkinsonism has been described after infection with several different viruses and it has therefore been hypothesized that a viral infection might play a role in the pathogenesis of the disease. We investigated formalin-fixed post-mortem brain tissue from 9 patients with Parkinson’s disease and 11 controls for the presence of Ljungan virus (LV) antigen using a polyclonal antibody against the capsid protein of this recently identified picornavirus with neurotropic properties, suspected of being both a human and an animal pathogen. Evidence of viral antigen was found in 7 out of 9 Parkinson’s disease cases and in only 1 out of 11 controls (*p* = 0.005). The picornavirus antigen was present in dopamine-containing neurons of the substantia nigra. We propose that LV or an LV-related virus initiates the pathological process underlying sporadic PD. LV-related picornavirus antigen has also been reported in patients with Alzheimer’s disease. Potentially successful antiviral treatment in Alzheimer’s disease suggests a similar treatment for Parkinson's disease. Amantadine, originally developed as an antiviral drug against influenza infection, has also been used for symptomatic treatment of patients with PD for more than 50 years and is still commonly used by neurologists today. The fact that amantadine also has an antiviral effect on picornaviruses opens the question of this drug being re-evaluated as potential PD therapy in combination with other antiviral compounds directed against picornaviruses.

## 1. Introduction

Parkinson’s disease (PD), amyotrophic lateral sclerosis (ALS), frontotemporal lobe degeneration (FTLD) and Alzheimer’s disease (AD) are all characterized by progressive degeneration and the loss of specific subsets of neurons, leading to decline in brain functions such as cognition and locomotor control [1]. Neurodegenerative diseases have distinct clinical manifestations depending on the localization of brain pathology, but also share many features such as protein aggregation and the formation of inclusion bodies [2].

Molecular epidemiological approaches have identified high-risk genes for familial neurodegenerative diseases and have also determined common genetic variants that may predict susceptibility for non-familial forms of these diseases [3,4]. A growing body of epidemiological and experimental data also points to the fact that chronic viral infections may cause neurodegenerative diseases [5]. In the case of PD, various genetic and environmental factors have also been associated with the disease and the number of known genetic risk factors has increased dramatically [6]. However, these known loci presently account for only approximately 20% of PD risk, meaning that a substantial proportion of PD remains unexplained on the bases of currently known genetic associations [6].

The idea that viral infections may initiate or cause PD often relates to findings from coincident cases that lie outside the expected norm. One of the most renowned examples is the parkinsonism that occurred subsequent to a viral encephalopathy that developed following the 1918 influenza pandemic [7,8,9]. In addition to that influenza pandemic there have been case reports of post-encephalitic parkinsonism following encephalitis associated with Japanese encephalitis virus, St Louis encephalitis virus, West Nile virus and picornavirus infection [9,10,11,12,13]. Laboratory mice infected with Western equine encephalitis virus have been found to develop activation of microglia and astrocytes, selective loss of dopaminergic neurons in the substantia nigra and neurobehavioral abnormalities [14]. The role of infectious etiology in neurodegenerative disease gained plausibility through the Braak hypothesis of pathological spread starting in the olfactory bulb and peripheral nerves of the gastrointestinal tract [15]. Either of these locations could be the entry site for an environmental trigger, whether toxic or infectious.

We recently reported the discovery of picornavirus antigen in formalin-fixed post-mortem brain tissue from the hippocampus region of AD cases, but not in age-matched controls, using immunohistochemistry (IHC) [16]. The goal of the present study was to extend our observation of picornavirus antigen in AD patients through investigating the presence of this same antigen in the substantia nigra of post-mortem PD patients.

## 2. Materials and Methods

### 2.1. Ethical Statement

All donors were registered by the Netherlands Brain Bank (NBB), Netherlands Institute for Neuroscience (Meibergdreef 47, 1105 BA Amsterdam, the Netherlands https://www.brainbank.nl/ accessed on: 9 January 2019). Ethical approval for the NBB procedures and forms was given by the Medical Ethics Committee of the Vanderbilt University Medical Center (Amsterdam, the Netherlands) (ref 2009/148). Procedures were conducted in accordance with the Code of Conduct for Brain Banking and the Declaration of Helsinki [17,18]. All donors provided informed consent for autopsy, storage and use of their tissue and anonymized clinical and neuropathological data for research purposes.

### 2.2. Specimens

Formalin-fixed paraffin-embedded brainstem tissue from patients with PD and control specimens was received from the NBB. All PD cases included had reported motor symptoms typical of PD. The diagnosis was confirmed by pathological findings characterized by a selective loss of neurons in the brainstem that produce the neurotransmitters dopamine and noradrenaline. Lewy bodies and Lewy neurites composed of abnormal aggregated proteins, primarily abnormal alpha-synuclein, were visible inside the affected nerve cells [19,20].

Controls were defined as cases with no clinical sign of Parkinson’s disease or dementia and no pathological findings suggesting Parkinson’s disease or other neurodegenerative disease in the brain at autopsy. The substantia nigra from 4 females and 5 males with PD was analyzed. The mean age of the 9 PD patients was 70 (range 57–78, median 72). The substantia nigra was analyzed from 11 control cases, 6 females and 5 males. The mean age of the controls was 68 (range 46–82, median 69).

### 2.3. Immunohistochemistry

Formalin-fixed paraffin-embedded brain tissue from the substantia nigra region was sectioned at 5 µm and analyzed for the presence of LV antigen using IHC as previously described, with minor modifications [21]. The presence of LV-specific antigen was visualized using a polyclonal serum raised in rabbits using a recombinant LV VP1 for immunization [22]. The IHC assay used Novocastra (RE7119) Epitope Retrieval Solution pH 9.0. As control we used serum from a rabbit immunized using the same protocol but with the carrier protein only. Tissues from LV-infected and non-infected animals were included as additional controls. The specificity of the rabbit antibodies was verified by analyzing control specimens generated by mixing infected tissue culture cells with non-infected cells followed by formalin fixation and paraffin embedding. The specificity of the reaction was also confirmed by blocking the signal with LV antigen in parallel with control antigen.

## 3. Results

### Picornavirus Antigen in Brain Tissue of Patients with Parkinson’s Disease

Picornavirus antigen was detected in 7 of 9 cases with PD and in 1 of the 11 healthy controls. The difference of viral presence in PD patients vs. controls was found to be significant (*p* = 0.005 Fisher’s exact two-sided test). In the PD cases, Lewy bodies and Lewy neurites composed of abnormal aggregated proteins, primarily abnormal alpha-synuclein, were visible inside the affected nerve cells. Figure 1A–D provides photomicrographs of substantia nigra tissue from PD patients, staining mostly pyknotic apoptotic remnants of pigmented neurons positive for viral antigen. Neuro-melanin-containing neurons stain brown while viral antigen stains red. Figure 2A,B illustrates a negative control with substantia nigra from a healthy control with neuro-melanin-containing neurons but no viral antigen.

## 4. Discussion

### 4.1. Evidence Suggesting a Chronic Persistent Picornavirus Infection in Parkinson’s Disease and Several Other Diseases of Unknown Etiology

Influenza virus, Japanese encephalitis virus, Western equine encephalitis virus, St Louis encephalitis virus, West Nile virus, human immunodeficiency virus, herpesvirus and picornavirus have all been reported to induce parkinsonism transiently as well as permanently [5,9,23,24,25]. We discovered a novel picornavirus (the Ljungan virus) as part of an outbreak investigation of lethal myocarditis among Swedish elite orienteers in the 1990s [26,27]. Ljungan virus (LV) was isolated from wild bank voles and the hypothesis that LV was causing lethal myocarditis in humans was mainly based on the association between cyclic wild rodent abundance and a similar cyclic variation in the incidence of lethal myocarditis in humans [28]. The finding that LV was related to the rodent-borne picornavirus, encephalomyocarditis virus (EMCV), prompted us to expand our search effort to include all diseases in the repertoire of this virus. EMCV was identified more than 70 years ago and was initially of particular interest because of its possible relationship with human neurological diseases such as poliomyelitis [29]. EMCV is still of general interest as a cause of myocarditis, encephalitis, diabetes, chronic inflammatory demyelinating disease resembling multiple sclerosis and hippocampal neural damage resulting in disrupted spatial memory in different species of animals [30,31]. However, there is still no definitive proof that EMCV is responsible for any neurodegenerative disease in humans. LV shows similarities to EMCV regarding disease repertoire through its association with diabetes in its natural wild rodent reservoir and in its ability to cause cerebral malformation in laboratory mice [32,33,34]. LV has also been associated with malformation in human fetuses, with intrauterine fetal death and sudden infant death, both in epidemiological studies and in studies demonstrating viral antigen in human cases, but not in controls [35,36,37]. In addition, LV is also genetically related to human picornavirus pathogens in the Parechovirus genus and its role as a human pathogen is therefore presently under investigation [38,39].

We recently reported the discovery of picornavirus antigen in brain tissue from AD patients. Formalin-fixed post-mortem brain tissues from the hippocampus region of the brain were analyzed by IHC using the same assay as in the present study. All AD cases included in this previous study had a verified clinical profile of AD and morphologically showed typical neuropathological changes scored as Braak 4–6 [16,40]. Control specimens came from patients with no history of cognitive impairment or suspected dementia syndrome and no post-mortem histopathological signs of neurodegenerative disease. Picornavirus antigen was detected in all 18 cases with AD but in none of the 11 age-matched controls. The picornavirus antigen was found in neurons and astrocytes. A distinct positive reaction was also detected in amyloid or neuritic plaques [16]. In this report we found evidence of viral antigen in the substantia nigra cells of PD patients significantly more often than in controls. In addition, viral antigen was detected in cells and structures involved in the pathogenesis of PD, such as neuro-melanin-containing neurons. It can be argued that an irrelevant viral infection may occur or increase in any organ or tissue as a result of insufficient viral defense when tissue damage occurs, regardless of the pathogenesis. However, to our knowledge there is no report in the medical literature of virus in significant amounts in cells involved in the disease pathogenesis and concurrently in negative controls that later proved to be insignificant.

### 4.2. Limitations of the Diagnostic Assay Used in the Present Study

The specificity of the antiserum used to identify the presence of viral antigen in tissue is crucial for the interpretation of the results in this study. It is well understood that a polyclonal antiserum raised against a specific virus may also react with related viruses in the same viral family, but we have not identified any related picornaviruses that show cross-reactivity with the antisera used here [22]. 

The standard method to confirm a picornavirus infection diagnosed by serology is PCR followed by sequencing of the amplified product. 

The fact that the viral antigen was detected by IHC only and not confirmed by techniques such as virus isolation or PCR is a limitation of the hypothesis proposed. We in fact made repeated efforts to confirm the IHC results by PCR using a variety of primer combinations based on available sequence information, and all attempts were unsuccessful (Niklasson et al., unpublished observations).

The development of consensus primers targeting the well-conserved areas of the 5′ untranslated region (5′UTR) is the most often used method for the detection of picornaviruses. However, it is impossible to predict the sequence variation of a novel virus based on serological information alone. Furthermore, the increased mutation rate within the 5′UTR region known to occur in persistent infections can cause additional problems in the search for useful 5′UTR PCR primers to confirm the presence of an unsequenced virus.

It is possible that the difficulties we experienced using PCR to confirm viral-antigen-positive specimens depends on strain variation within the LV clade, and that only a small number of isolates have been fully sequenced. In this context we posit that an as-yet unidentified LV or LV-related virus could be antigenically close enough to be detected by an assay based on a polyclonal antiserum, but genetically distant enough to avoid recognition by PCR primers designed using sequence information from the few LV isolates sequenced to date [41,42].

### 4.3. Evidence of Persistent Picornavirus Infection

Multiple studies have shown that various different picornaviruses are able to produce persistent intracellular low-replicative steady-state infections without acute cytopathic effects and cell death. Signs of persistent picornavirus infection were reported in the 1980s when viral RNA was detected in the majority of heart tissue biopsies analyzed from myocarditis or dilated cardiomyopathy patients, whereas all biopsies from controls with non-viral heart diseases were negative [43,44]. Evidence of persistent picornavirus infection has also been found in several other chronic diseases, including idiopathic dilated cardiomyopathy, chronic inflammatory myopathy, post-polio syndrome and type 1 diabetes (T1D) [45,46,47,48,49,50,51,52]. Chronic non-cytolytic picornavirus infection has been shown to induce autoantibodies which target mitochondria and thus substantially inhibit energy metabolism [53,54,55,56,57,58]. At the same time this chronic infection results in an altered immune response, making it increasingly difficult for the immune system to eradicate the infection [59]. The reason why the infection can evade the immune response is not clear [46,60,61,62,63]. In normal lytic picornavirus infections, there is around 100 times more positive-strand RNA than negative-strand RNA in infected cells. However, in the persistently infected tissues of animals and humans, roughly equal amounts of positive- and negative-strand RNA were found, suggesting that a mutant defective virus may be responsible for these persistent muscle infections [53,64,65,66,67]. In addition, the total tissue viral load in patients with persistent picornavirus infection has been found to be very low, posing a significant challenge when trying to detect the virus using PCR [63]. Persisting non-cytolytic picornaviruses with deletions in the 5′ region have also been found in the pancreas of laboratory rodents [49,68].

Alpha-synuclein is a protein expressed uniquely in neurons. Aggregation of alpha-synuclein is the primary component of Lewy bodies that directly correlates with PD. The role of the neuronal expression of alpha-synuclein is still unknown, but there is data suggesting alpha-synuclein expression in neurons is a restriction factor, inhibiting viral replication in the CNS [69]. In contrast, a recent study demonstrated that alpha-synuclein enhanced picornavirus replication in an experimentally infected PD mouse model, resulting in lower survival rate [70].

### 4.4. The Potential Role of Antiviral Treatment in Parkinson’s Disease and Several Other Chronic Diseases of Unknown Etiology

The BioBreeding (BB) rat is one of the most utilized animal models for the study of T1D [71]. The model originates from a colony of outbred Wistar rats that spontaneously developed hyperglycemia and fatal diabetic ketoacidosis. The onset of symptomatic diabetes in BB rats has a predictable and narrow time frame, providing an excellent opportunity to study the impact of treatment before, during or after disease onset. We previously reported the observation that LV antigen is present in the islets of Langerhans in BB rats at the time of diabetes onset [72]. This observation was followed by experiments to see if diabetes onset could be postponed, stopped or reversed using antiviral compounds. We investigated antiviral compounds alone and in combination focusing on drugs with reported effect on picornaviruses and that have already been approved for human use. Among the drugs tested were Pleconaril, a compound selectively inhibiting picornavirus replication, and ribavirin, a nucleotide analogue of guanosine with a broad antiviral spectrum including effect on members of the picornavirus family.

Efavirenz, widely used in antiviral therapy for human immunodeficiency virus infection, was also found to have an antiviral effect in picornavirus infections. This was unexpected, since picornavirus does not use reverse transcriptase for its replication. This suggests that efavirenz has an additional antiviral mechanism which has not yet been described [73,74]. We tested these drugs alone and in combination, finding that a single antiviral compound resulted in measurable delay in age of onset of T1D disease, whereas a combination of two or three antiviral compounds could prevent the onset of T1D altogether [73]. Combinations of antiviral therapy given after T1D onset could also reverse symptoms of diabetes in conjunction with the disappearance of viral antigen from the insulin-producing β-cells [73].

In another study, three patients diagnosed with AD received combinations of pleconaril, ribavirin and efavirenz during a time period of 4–8 years, according to the international guidelines current at the time. The cognitive status of these three patients (female 81, male 79 and female 68 years of age) was followed with Mini Mental State Examination (MMSE) score and Alzheimer’s Quick Test (AQT) [75,76]. Both these instruments are well-established and validated tools for the evaluation of cognitive function in AD. The MMSE has a maximum score of 30 points and decreases an average of 3 points per year as mental function decreases. The AQT is measured in seconds to completion, and typically increases by 16 s per year [75,76].

If you introduce treatment targeting a disease with a known decline rate, the effect of the therapy (if any) can be expressed as decreased decline rate (DDR). DDR is used to characterize the difference between the decline rate in historical controls (a statistical population) and the clinical change seen in an investigated patient. The DDR as a percentage is calculated using the formula DDR = (1 − (actual change/expected change)) × 100. Full treatment effect without decline is expressed as DDR 100%. If patients present a reduced decline rate as a result of treatment, the DDR will be between 1% and 100%, while improvement is expressed as DDR >100%. For patients declining faster than expected, the DDR will show a negative outcome.

The DDRs for the MMSE were 96%, >100% and >100% for patients 1, 2 and 3, respectively. The corresponding figures for the AQT were >100%, 89% and >100% for patients 1, 2 and 3, respectively. This translates to more or less unchanged cognitive function in two patients and a clear improvement in one patient during the treatment period, which is in contrast with the expected disease progression of AD. In addition, there were observations of cognitive decline in all three patients when antiviral therapy was interrupted for different reasons. The recovery of lost capacity seen after reinstatement of therapy suggests a direct causal impact of antiviral compounds on a viral infection in the brain (Niklasson B, Lindquist L and Klitz W, unpublished observation).

Based on the previously reported findings of picornavirus antigen in brain tissue from AD patients and a positive clinical effect observed in a small number of patients receiving antiviral treatment as part of regular health care, a double-blinded, placebo-controlled study using a combination of pleconaril and ribavirin was performed at the Karolinska Institute Stockholm, Sweden, sponsored by Apodemus AB [77]. Patients in this clinical trial received antiviral therapy or a placebo for 9 months, after which they were tracked for an additional 12 months. Cognitive function was measured before medication and at 3, 6 and 9 months during medication and at 1, 6 and 12 months following termination of therapy. When cognitive function for each patient was compared with their status prior to starting treatment, the patient group receiving the placebo decreased in cognitive function over time as expected. The patient group receiving active treatment showed continuous improvement until one month after termination of therapy when the curve changed direction and cognitive function started decreasing following a similar negative rate of change to that seen in the placebo group. The group receiving antiviral therapy had better cognitive function compared to the placebo group each time cognitive function was measured during the clinical trial. The positive difference between patients receiving active therapy compared to placebo controls was statistically significant at 1 and 12 months after termination of therapy [77].

Amantadine was originally developed as an antiviral compound for influenza A. It was accidentally discovered that amantadine also had a positive effect on PD in PD patients receiving the drug when suffering from an influenza infection. Amantadine was initially found to increase the amount of dopamine released in the brain in PD patients. This was long believed to be a direct effect of the drug [78]. The finding that amantadine also has antiviral effects against picornavirus opens the possibility that the effect seen may indeed be due, at least in part, to its antiviral capacity [79].

## 5. Conclusions

The goal of this study was to extend our previous reported findings of picornavirus antigen in the brains of AD patients and to investigate whether viral antigen could be detected in PD patients more often than in controls. The discovery of LV antigen in the substantia nigra cells of PD patients significantly more often than in controls raises the possibility that the same viral antigen can be found in many neurodegenerative diseases. This opens the hypothesis of a common viral infection in neurodegenerative disease taken as a whole. We believe that patients with FTLD and ALS and corresponding controls should also be investigated.

While a more definitive confirmation of viral involvement in PD is necessary and crucial, it is possible to evaluate the clinical effect of immediate initiation of antiviral PD treatment by assessing the development of disease symptoms using methods for measuring DDR. Combinations of antiviral therapies have become well established for the treatment of many viral infections due to better efficacy, reduced toxicity and an ability to prevent the development of viral resistance [80].

The general advantage of combination therapy over monotherapy is supported by theoretical models of virus dynamics and by treatment experience [81]. Pleconaril, efavirenz, amantadine, ribavirin and fluoxetine are among drugs with antiviral effects on picornaviruses which are available for human use [82,83]. The establishment of a positive impact of antiviral medication on PD would offer proof of concept regarding viral involvement in the pathogenesis of PD and simultaneously offer a new primary pathway to PD treatment.

This would lead to further important studies to identify the responsible virus, and opens the way for an explanation of the long-standing difficulties in retracing the pathogenic chain in PD.

## Figures and Tables

**Figure 1 microorganisms-10-00599-f001:**
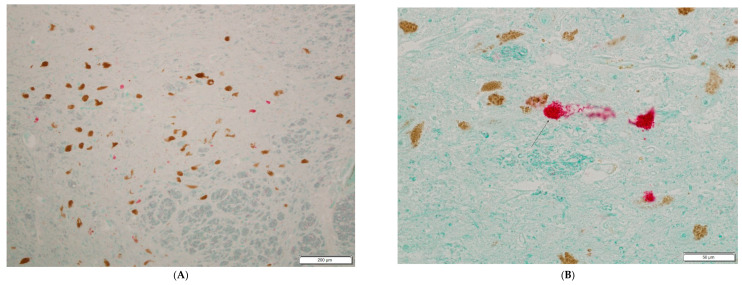
Neuro-melanin-containing neurons in the substantia nigra undergo neurodegeneration during Parkinson’s disease. (**A**) illustrates an overview of the substantia nigra in a patient with PD while (**B**–**D**) are magnifications focusing on one cell indicated by an arrow, staining positive for both neuro-melanin (brown) and LV VP1 (red).

**Figure 2 microorganisms-10-00599-f002:**
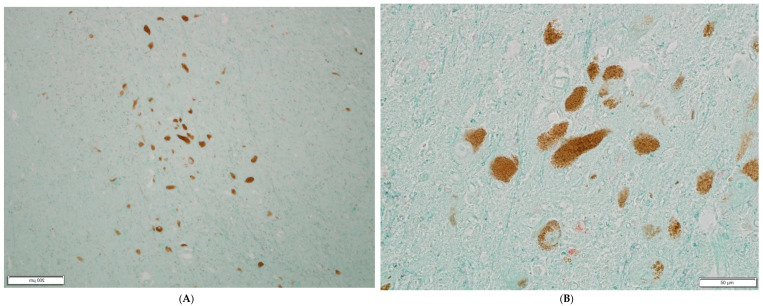
(**A**,**B**) illustrate an overview of the substantia nigra from a healthy control patient.

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
