# Peer review of "Picornavirus May Be Linked to Parkinson’s Disease through Viral Antigen in Dopamine-Containing Neurons of Substantia Nigra"

_microorganisms, 2022, doi:10.3390/microorganisms10030599_

Round 1

Reviewer 1 Report

The manuscript " Picornavirus may be linked to Parkinson´s disease through viral antigen in dopamine-containing neurons of substantia nigra" can be accepted for publication in the microorganisms after minor revision. This research was interesting.

Comment #1- Please improve conclusion.

Comment #2- The English language should be corrected carefully. The text should be checked for grammatical errors, singular/plural, and long sentences throughout the whole manuscript. There are numerous mistakes in the text!

Comment #3- Please use new references.

Author Response

Comment 1. Text section 4.5 Conclusions has been re-written as suggested

Comment 2. The manuscript has been checked again for grammatical errors

Comment 3.  We agree with the reviewer that many of the citations are old. We have added 5 recent citations (12, 13, 14, 69 and 70). The hypothesis of viral involvement in several chronic diseases including neurodegenerative diseases was presented long ago. As a consequence of this many citations were published years ago and sometimes even decades ago. 

Text that has been changed in response to suggestions made by reviewer 1-3 is highlighted in yellow

Reviewer 2 Report

This is an important study and piece of work by the authors in correlating picronavirus infection wit Parkinson's disease. I have 2 major comments on this work:

1. This information is of extreme importance. However, it must be studied further that Parkinson's disease is related to the infection or the viral antingen undergoes aggregation thereby providing a seed for for alpha synuclein to aggregate. May be some in vitro experiments for the aggregation of alpha synuclein in the presence of Ljungan virus antigen can be carried out to test this hypothesis.

2. Some orthogonal methods should be utilized to correlate with the immunohistochemistry findings. Such methods may include mass spectrometry, western blotting, co-IP etc.

Author Response

We agree with the reviewer that experiments to understand how a viral infection can cause or interplay with the development of alpha synuclein aggregates is very important. We have added a short paragraph in the discussion regarding alpha synuclein and also added two recent citations (69-70).  We are aware of the fact that our report raises several questions and that we leave a number of them unanswered. It would also be of great interest to follow detectable virus antigen and alpha synuclein over time in specimens collected at different stages of the disease.

We hope that the present report suggesting an association between a virus and PD will result in research efforts to investigate how this  virus found at the “crime scene” cause the disease we define as PD. We also hope that our observations, without delay, result in efforts to treat patients with PD using antiviral compounds already available. There are many examples in the history infectious disease were research focused on therapy have resulted both in effective therapeutic measures for the disease in question and also turned out to be a useful tool studying the pathogenesis of the disease.  Text that has been changed in response to suggestions made by reviewer 

Reviewer 3 Report

This study represents an interesting contribution to the field, has scientific quality and also very well written.

Nevertheless, some minor considerations have to be clarified or fixed:

The novelty character of this paper should be better marked. Please rewrite the “goal” of the present study?  It's not clear the main message of the study.

Data and that deal with Picornavirus and Parkinson's disease should be entered in the introduction.

In general, the "discussion" section should be more substantiated to highlight your results. This is a well-written theoretical chapter, but with minimal references to your results.

The literature is quite «old» and insufficient. Please add literature of the last five years. Generally, the literature is quite «old» and insufficient. Please add literature of the last five years. If it does not exist, its lack should be emphasized and the fact that your research comes to fill a gap.

Please discuss the methodological limitations of the article. A «limitation» section should be structured.

Please complete subsection “Conclusion” with information on potential practical application of the obtained results of this study.

Author Response

Text regarding the goal of the study has been included both in the Introduction and in the Discussion. The last part of the introduction has been re-written to explain the hypothesis of picornavirus and PD.

We agree with the reviewer that many of the citations are old. We have added 5 recent citations (12, 13, 14, 69 and 70). The hypothesis of viral involvement in several chronic diseases including neurodegenerative diseases was presented long ago. As a consequence of this many citations were published years ago and sometimes even decades ago. 

The limitations of the present study have been emphasized by introducing a separate headline with the title “Limitations of the diagnostic assay used in the present study” in the Discussion.

Regarding practical applications of the present report we believe that we have been quite specific under the headline Conclusions.

Text that has been changed in response to suggestions made by reviewer 

This manuscript is a resubmission of an earlier submission. The following is a list of the peer review reports and author responses from that submission.